# Diagnostic Performance of Publicly Available Large Language Models in Corneal Diseases: A Comparison with Human Specialists

**DOI:** 10.3390/diagnostics15101221

**Published:** 2025-05-13

**Authors:** Cheng Jiao, Erik Rosas, Hassan Asadigandomani, Mohammad Delsoz, Yeganeh Madadi, Hina Raja, Wuqaas M. Munir, Brendan Tamm, Shiva Mehravaran, Ali R. Djalilian, Siamak Yousefi, Mohammad Soleimani

**Affiliations:** 1Department of Ophthalmology, University of North Carolina at Chapel Hill, Chapel Hill, NC 27599, USA; cheng.jiao@unchealth.unc.edu (C.J.); erikrosas.10@gmail.com (E.R.); 2Department of Ophthalmology, University of California San Francisco, San Francisco, CA 94143, USA; hassan.asadigandomani@ucsf.edu; 3Department of Ophthalmology, Hamilton Eye Institute, University of Tennessee Health Science Center, Memphis, TN 38103, USA; delsoz_mohammad@yahoo.com (M.D.); yeganeh.madadi@gmail.com (Y.M.); hinaraja65@gmail.com (H.R.); siamak.yousefi@uthsc.edu (S.Y.); 4Department of Ophthalmology and Visual Sciences, University of Maryland School of Medicine, Baltimore, MD 21201, USA; wmunir@som.umaryland.edu (W.M.M.); btamm@som.umaryland.edu (B.T.); 5Department of Biology, School of Computer, Mathematical, and Natural Sciences, Morgan State University, Baltimore, MD 21251, USA; shiva.mehravaran@morgan.edu; 6Department of Ophthalmology and Visual Sciences, University of Illinois at Chicago, Chicago, IL 60612, USA; adjalili@uic.edu; 7Department of Genetics, Genomics, and Informatics, University of Tennessee Health Science Center, Memphis, TN 38136, USA

**Keywords:** ChatGPT, large language models, corneal disease, artificial intelligence

## Abstract

**Background/Objectives:** This study evaluated the diagnostic accuracy of seven publicly available large language models (LLMs)—GPT-3.5, GPT-4.o Mini, GPT-4.o, Gemini 1.5 Flash, Claude 3.5 Sonnet, Grok3, and DeepSeek R1—in diagnosing corneal diseases, comparing their performance to human specialists. **Methods:** Twenty corneal disease cases from the University of Iowa’s EyeRounds were presented to each LLM. Diagnostic accuracy was determined by comparing LLM-generated diagnoses to the confirmed case diagnoses. Four human cornea specialists evaluated the same cases to establish a benchmark and assess interobserver agreement. **Results:** Diagnostic accuracy varied significantly among LLMs (*p* = 0.001). GPT-4.o achieved the highest accuracy (80.0%), followed by Claude 3.5 Sonnet and Grok3 (70.0%), DeepSeek R1 (65.0%), GPT-3.5 (60.0%), GPT-4.o Mini (55.0%), and Gemini 1.5 Flash (30.0%). Human experts averaged 92.5% accuracy, outperforming all LLMs (*p* < 0.001, Cohen’s d = −1.314). GPT-4.o showed no significant difference from human consensus (*p* = 0.250, κ = 0.348), while Claude and Grok3 showed fair agreement (κ = 0.219). DeepSeek R1 also performed reasonably (κ = 0.178), although not significantly. **Conclusions:** Among the evaluated LLMs, GPT-4.o, Claude 3.5 Sonnet, Grok3, and DeepSeek R1 demonstrated promising diagnostic accuracy, with GPT-4.o most closely matching human performance. However, performance remained inconsistent, especially in complex cases. LLMs may offer value as diagnostic support tools, but human expertise remains indispensable for clinical decision-making.

## 1. Introduction

Corneal disease encompasses a diverse range of conditions affecting the transparent surface of the eye, significantly affecting vision and quality of life. Corneal opacities are a leading cause of blindness worldwide, affecting an estimated 5.5 million individuals [1]. However, diagnosis and treatment are often limited by the availability of ophthalmologists, as well as socioeconomic barriers to care [2]. Consequently, despite advancements in medical care, the burden of preventable blindness continues grow alongside global population expansion and aging, further exacerbated by a shortage of specialists [3]. Artificial intelligence (AI) presents promising solutions for disease triage and diagnosis. Initial AI application in ophthalmology focused primarily on the posterior segment, addressing conditions such as diabetic retinopathy, macular degeneration, and glaucoma [4,5,6]. This led to the development and FDA approval of IDX-DR, the first AI-based diagnostic tools for detecting diabetic retinopathy [7]. In comparison, progress in AI development for the anterior segment has occurred at a slower rate. Nonetheless, there have been prior efforts to achieve the automatic detection and classification of features of microbial keratitis, keratoconus, dry eye syndrome, and Fuchs endothelial dystrophy [8,9,10,11,12,13,14,15,16,17].

Large language models (LLMs) are a type of AI designed to understand and generate human language [18]. These models are trained on massive collections of text data and use deep learning algorithms to predict and construct coherent responses based on input prompts [18]. Unlike traditional rule-based systems, LLMs are capable of contextual understanding, logical reasoning, and dynamic adaptation across a wide range of topics [18]. Their flexibility and general-purpose capabilities have led to their rapid integration into various fields, including medicine, where they hold promise for supporting clinical decision-making and diagnosis [19].

Recent advancements in LLMs, a subset of AI deep learning, have captured worldwide attention for their user-friendly interfaces and general-purpose intelligence [18]. These models represent a significant development in the AI field as they are capable of processing and generating human-like output while also understanding contextual information and emulating logical reasoning to generate accurate conclusions in the fields of medicine and computing [18]. Within ophthalmology, ChatGPT (January 9 “legacy” and ChatGPT Plus), (OpenAI, San Francisco, CA, USA) demonstrated promising performance, scoring 55.8% on the Ophthalmic Knowledge Assessment Program (OKAP) [20]. A subsequent iteration of the GPT model showed substantial improvements, reaching up to 82.4% accuracy, exceeding the 75.7% of the human respondents to the American Academy of Ophthalmology (AAO) Basic and Clinical Science Course (BCSC) self-assessment program [21]. Similarly within corneal diseases, ChatGPT has shown great promise, with diagnostic accuracy improvements from 60% to 85% for GPT-3.5 and GPT 4.0, respectively [22].

Since then, significant advancements have been achieved, with the release of multiple new sophisticated LLMs, including GPT-4.o, with the “o” representing “omni” (released May 2024; OpenAI, San Francisco, CA, USA), Gemini 1.5 (released May 2024; Google, Mountain View, CA, USA), and Claude Sonnet 3.5 (released October 2024; Anthropic, San Francisco, CA, USA). These newer models represent an evolution in LLM architecture and training, incorporating several key advancements. Firstly, multimodal capabilities have emerged, enabling models to process and integrate information from different sources. For instance, Gemini was trained on comprehensive datasets encompassing web text, code repositories, Google internal data (Search, Books, YouTube), and publicly available datasets, facilitating a more holistic understanding even of standard text-based inputs [23]. Secondly, expanded context windows represent a crucial improvement in handling longer and more complex inputs. As compared to the prior GPT 3.5 4 k tokens, GPT 4.o has an expanded context window of 128 k (approximately 300 pages of text), while that of Google’s Gemini 1 is million [24]. Finally, these newer models benefit from more curated and diverse training datasets. While earlier LLMs are often trained primarily from large corpora of text from the internet, newer models leverage more curated datasets. Claude’s approach with the Anthropic library involves curating materials and using “constitutional AI” training that is fine-tuned to avoid harmful, biased, or misleading outputs [25]. Also, two additional models—Grok3 (xAI, released June 2024) and DeepSeek R1 (DeepSeek, released July 2024)—have entered the landscape, also offering publicly accessible platforms with competitive performance and multimodal capabilities. Grok3 emphasizes real-time reasoning and open-domain adaptability, while DeepSeek R1 integrates curated medical sources into its training pipeline.

This rapid progress in LLM architectures and technical advancements necessitates a comprehensive evaluation of their capabilities, particularly in specialized domains such as corneal disease. This study aimed to assess the diagnostic accuracy of seven publicly available LLMs—GPT-4.o, Claude 3.5 Sonnet, GPT-3.5, GPT-4.o Mini, Gemini 1.5 Flash, Grok3, and DeepSeek R1—in identifying corneal eye diseases and to compare this accuracy with that of human corneal specialists. Given the increasing prevalence of corneal diseases and persistent challenges of access to specialized care, the evaluation of innovative tools will hopefully improve diagnostic efficiency.

## 2. Materials and Methods

### 2.1. Case Collection

The cases used for this study were collected from EyeRounds, the University of Iowa’s Department of Ophthalmology and Visual Sciences Ophthalmology Cases [26]. These case reports are free to access and available to the public. Twenty-cases from the Cornea/External Eye Disease category were selected. Patients requiring specialized diagnostic evaluations, such as those with Fusarium fungal keratitis, were excluded from selection. The following case reports were selected for the study: acanthamoeba keratitis, acute corneal hydrops, atopic keratoconjunctivitis, calcific band keratopathy, Cogan’s syndrome, corneal marginal ulcer, cystinosis, cytarabine-induced keratoconjunctivitis, exposure keratopathy, Fabry disease, Fuch’s endothelial corneal dystrophy (FECD), herpes simplex virus keratitis, infectious crystalline keratopathy, lattice corneal dystrophy type II, megalocornea, peripheral ulcerative keratitis, posterior polymorphous corneal dystrophy (PPCD), pseudophakic bullous keratopathy (PBK), Salzmann’s nodular degeneration (SND), and amiodarone-induced corneal deposits. This retrospective observation study analyzed case reports that contained anonymized patient information, including demographics, chief complaints, presenting symptoms, and major exam findings. A standardized pre-processing approach was adopted to transform the comprehensive ophthalmological case report into concise clinical vignettes. Through use of case materials from the Iowa EyeRounds database, essential clinical information was extracted while key diagnostic elements such as patient demographics, chief complaint, relevant history, and clinical exam findings were preserved. This standardization ensured consistency across all evaluated LLMs while maintaining the diagnostic integrity and clinical challenge of each case. Pre-processing eliminated educational content such as treatment discussion, pathophysiology explanation, and literature references that would not typically be available during diagnostic assessment. All 20 selected cases are included in the Appendix A as a PDF file (Appendix A). Since no patient-identifiable information was utilized and the dataset was publicly available, our institution determined that no formal institutional review board approval was necessary. This study adhered to the principles of the Declaration of Helsinki. Table 1 shows the distribution of the corneal disease cases based on disease categories (congenital, degenerative, infectious, and inflammatory) and patient demographics (Table 1).

### 2.2. Large Language Model Selection

The LLMs used for this study included GPT 3.5, GPT 4.o mini, Chat GPT 4.o, Google Gemini 1.5 flash, Grok3, DeepSeek R1, and Claude Sonnet 3.5. These seven models were chosen primarily for their public accessibility, which eliminates the need for specialized training, expertise, or additional costs—making them more practical for most healthcare practices [27]. For instance, GPT 4.o mini can be directly accessed through web search, without even an email registration. The only LLMs that required paid subscription were GPT 4.o and DeepSeek R1.

The model selection targeted the leading commercial LLMs from major AI developers (DeepSeek, xAI, OpenAI, Google, and Anthropic) that represented the state-of-the art at the time of study initiation. Importantly, all selected models featured training with multimodal data, allowing them to process both text and image inputs—a critical requirement for comprehensive ophthalmology assessments. This multimodal integration represents a significant advancement over text-only LLMs (such as GPT-3.0) and is particularly valuable in ophthalmology, where diagnosis depends on visual interpretation. Corneal practice relies heavily on slit-lamp examination and corneal topography. Thus, models trained on diverse image datasets theoretically possess enhanced capabilities for interpreting cornel pathology.

These models differ in their training data and approach. OpenAI’s GPT models were trained using their massive internal library and initially built for general language task. In contrast, Grok3 (developed by xAI) leverages real-time web data through X (formerly “Twitter”) and emphasizes reasoning and factual alignment. DeepSeek R1 incorporates both English and Chinese medical content using a hybrid retrieval-augmented generation (RAG) pipeline to enhance factuality. In comparison, Gemini was trained using Google’s vast resource collection of search data, publications, and code repositories for a multimodal content of text, code, audio, and image data [23]. Claude prioritizes quality and safety over sheer volume by using Anthropic’s curated library [25].

### 2.3. Diagnosis

Each cornea case from the Iowa EyeRounds was individually entered into each of the AI models. The summarized clinical vignettes were presented with a final prompt for each input such as “What is the most likely diagnosis?” No further prompt engineering was introduced. To ensure consistent and unbiased results, we utilized new accounts with no prior usage for each model. We also cleared the conversation history after each response to prevent bias from previous interactions. If the model’s response was ambiguous, a re-prompt was issued. For instance, on occasion, the LLM response would suggest seeking medical care with an ophthalmologist and not give a formal diagnosis. These LLM responses would then be re-prompted with “Provide the diagnosis”. The accuracy of each model’s responses was determined by comparing its output diagnosis to the known formal diagnosis within the EyeRound case.

The study did not include specialized prompting techniques or custom-trained LLMs as the results aimed to establish baseline performance levels. This approach reflects the typical experience of average end-users in clinical settings who may lack expertise in prompt engineering. Additionally, custom implementation introduces significant methodological variability, as each model’s fine-tuning process differs substantially in approach and optimization techniques, potentially confounding comparative analyses.

Each of the cornea specialists received the same cases that were input into the LLMs. Diagnostic accuracy was determined by comparing each LLM’s and human expert’s diagnosis against the confirmed clinical diagnosis for each original corneal case. A correct response was scored as 1 and an incorrect response as 0. The overall accuracy was calculated as the percentage of correct diagnoses out the total number of cases for each diagnostician. A human consensus diagnosis was established using a majority rule approach, where agreement among three or more specialists (≥3/4) was considered consensus.

Statistical analyses were performed using SPSS (Version 26, IBM corp., Armonk, NY, USA) and Excel (Microsoft, Redmond, Washington, DC, USA). The diagnostic accuracy for both human specialist and LLMs was calculated. To compare performance between LLMs, Cochran’s Q test was employed for multiple related samples, followed by post hoc pairwise comparisons using McNemar tests with Bonferroni correction. The overall comparison between LLM and corneal expert performance utilized paired t-tests, comparing the average LLM score against the average human score. Effect sizes were calculated using Cohen’s d with Hedges’ correction. Interobserver agreement among LLM and human specialist was assessed using the number of matched diagnostic response between the two groups, as well as Cohen’s kappa coefficient for all possible pairwise combinations of human experts. The disease-specific analyses stratified case by condition type (congenital, degenerative, infectious, and inflammatory) (Figure 1).

## 3. Results

The diagnostic accuracy of the evaluated language models varied considerably (Table 2). GPT-4.o achieved the highest accuracy at 80.0% (16/20 cases), followed by Claude 3.5 Sonnet and Grok3 at 70.0% (14/20). DeepSeek R1 achieved an accuracy of 65.0% (13/20), GPT-3.5 achieved 60.0% (12/20), GPT-4.o Mini achieved 55.0% (11/20), and Google Gemini 1.5 Flash demonstrated the lowest accuracy at 30.0% (6/20) (Figure 2). The LLMs achieved an average accuracy of 61.4%. Cochran’s Q test among the LLMs showed significant variability within the group (*p* = 0.001). The four cornea specialists (H1, H2, H3, and H4) served as the human expert comparison group. Their diagnostic accuracy was 100.0%, 90.0%, 90.0%, and 90.0% respectively. The overall human expert average was 92.5%, with the human expert consensus (agreement among ≥3/4 experts) achieving 90.5%. There was no statistically significant difference in diagnostic accuracy among the four human experts (*p* = 0.522), indicating high consistency in their diagnostic capabilities. Human experts significantly outperformed LLMs in diagnostic accuracy (92.5% vs. 61.4%; *p* < 0.001). The mean difference was −0.311 (SE = 0.059, 95% CI [−0.435, −0.186]). The effect size was large (Cohen’s d = −1.168; Hedges’ correction = −1.121, 95%), indicating a substantial performance gap.

There was considerable variability in interobserver agreement with the cornea specialists across the LLMs (Table 3). GPT 4.o achieved the highest agreement with the specialists (80%, 75%, 85%, and 80%). Claude demonstrated moderate agreement at 60% with each the specialist. Both GPT-4.o mini (55%, 50%, 60%, and 65%) and GPT-3.5 (all 60%) showed similar agreement, while Gemini exhibited the lowest at 30%. Notably, GPT-4.o was the only LLM that showed significant agreement with human consensus (Kappa = 0.348, *p* = 0.040). Claude 3.5 Sonnet demonstrated fair but non-significant agreement with human consensus (*p* = 0.117). DeepSeek R1 and Grok3 both exhibited moderate agreement with human specialists, with interobserver agreement scores ranging from 60% to 75%. The Kappa values for agreement with the human consensus were 0.178 (*p* = 0.162) for DeepSeek R1 and 0.219 (*p* = 0.117) for Grok3, indicating fair but non-significant agreement (Table 3).

There were significant differences between the LLMs and human experts. Both GPT 4.o mini and Gemini 1.5 Flash demonstrated significant differences from human consensus (*p* = 0.008 & *p* < 0.001) and all four individual experts. GPT-3.5 showed significant difference from human consensus (*p* = 0.016) and only three individual experts (H1: *p* = 0.008; H2: *p* = 0.031; H4: *p* = 0.031), with these differences primarily reflecting cases where humans were correct and the model incorrect. Claude 3.5 Sonnet showed mixed results, with a marginally significant difference from human consensus (*p* = 0.063) and a significant difference from one expert (H1: *p* = 0.031). Notably, GPT-4.o demonstrated no significant differences from human consensus (*p* = 0.250) or any individual expert (*p*-values ranging from 0.125 to 0.625), suggesting performance comparable to human diagnosticians.

Pairwise comparisons between LLMs showed that GPT-3.5 significantly outperformed Gemini 1.5 Flash (*p* = 0.031), as did GPT-4.o (*p* = 0.002) and Claude 3.5 Sonnet (*p* = 0.021). A marginally significant difference was observed between GPT-4.o mini and GPT-4.o (*p* = 0.063). No significant differences were found between other LLM pairs, including GPT-4.o vs. Claude 3.5 Sonnet (*p* = 0.687), suggesting comparable performance between these two top-performing models.

Analysis by corneal disease category (as shown in Table 1) showed GPT-4.o achieved perfect accuracy in both degenerative (6/6) and infectious conditions (3/3). All models struggled with inflammatory cases, with accuracy ranging from 0 to 60%. Most models achieved 66.7% (2/3) accuracy with infectious cases. There was considerable spread in performance, with congenital cases ranging from 33.3% (2/6) to 83.3% (5/6). Notably, DeepSeek R1 also achieved 100% accuracy in degenerative cases and 66.7% in infectious cases, while Grok3 matched GPT-4.o achieved 100% accuracy in infectious cases and demonstrated strong performance in congenital cases at 66.7% (Table 4).

## 4. Discussion

This study prospectively evaluated the diagnostic accuracy of five publicly available AI language models—GPT 3.5, GPT 4.o mini, GPT 4.o, Gemini 1.5 Flash, and Claude 3.5 Sonnet—in diagnosing 20 cases of various corneal eye diseases, comparing their performance to that of human corneal specialists. GPT 4.o demonstrated the highest diagnostic accuracy among the evaluated LLMs, followed by Claude 3.5 Sonnet and Grok3. Gemini exhibited the lowest accuracy, performing significantly worse than all human experts, while GPT 4.o, Grok3, and Claude’s performance was comparable to that of the specialists. Interobserver agreements with specialists were strongest with GPT 4.o and weakest with Gemini. A summary of prior important studies evaluating LLM performance in ophthalmology is presented in Table 5 to provide broader context and comparison with our findings (Table 5).

Regarding prior research, GPT 3.5 had correctly answered 58% of the multiple choice OKAP practice questions, while GPT-4.0 (prior model to 4.o) accuracy improved to 84%. When analyzed by specialty, cornea was one of the strongest areas, with an accuracy of 90% [35]. Another study based on AAO clinical cases showed that GPT 3.5 correctly answered 71% of the multiple choice cornea and anterior segment questions, whereas GPT-4.0 answered 87% correctly [36]. Similarly, Google’s Gemini 1.0 had an accuracy of 71% for cornea and external disease [29]. More relevant to our study, ChatGPT 4.0 was able to correctly identify the diagnosis in 88.2% of cases in an ophthalmic clinical cases series [28]. Within our study, GPT 4.o maintained a similar degree of accuracy as that of GPT 4.0 at 80%. However, GPT 4.o mini had accuracy that was more akin to the older GPT 3.5 model. Interestingly, Claude 3.5 Sonnet was able to maintain similar accuracy to that of the latest GPT model (70% vs. 80%), with similar agreement with human experts. Unlike prior studies, Google Gemini vastly under-performed when presented with corneal clinical cases as compared to multiple-choice general ophthalmology questions [29]. DeepSeek-R1’s performance in our study aligns with these findings, achieving a 65% accuracy in corneal disease diagnosis, which is comparable to Claude 3.5 Sonnet’s performance. Grok 3’s accuracy of 70% also placed it among the top-performing models, surpassing several others in our evaluation. Our study, which focused on challenging corneal cases, reaffirms that various new AI systems are rapidly progressing but still struggle to master complex, specialized real-world tasks beyond test materials.

Recent advancements in LLMs mark a significant improvement in ophthalmology performance, which has been driven by key technical innovations. Unlike traditional machine learning, which relies heavily on pre-labeled data tailored for specific tasks, modern transformer-based models, such as GPT, Claude, and Gemini, are capable of zero-shot learning [37]. This paradigm enables these models to perform new tasks by leveraging semantic relationships and patterns identified in prior data, without requiring explicit task-specific training [38]. A critical improvement lies in the training data and architecture. Both GPT-4 and Google Gemini utilize vast and diverse datasets, incorporating multimodal information from text, audio, and video [23,24]. This approach allows the models to operate across various domains, making them more adaptable to different tasks. While OpenAI has explicitly chosen to not disclose the internal architecture of GPT-4 in their technical report [39], Google’s Gemini leverages its Pathways architecture. Unlike traditional methods that activate all neurons for every task, Pathways utilizes sparse activation, where only a tailored subset of neuron is activated for specific inquiries, allowing for efficient scaling without a proportional increase in computational demand. Anthropic’s Claude model emphasizes post-training refinement using reinforced learning from human feedback (RLHF) [40]. This approach involves a human reviewer assessing model outputs and helping the model align with desired behaviors and reducing the frequency of “hallucinations” (instances where the model generates incorrect yet confident-sounding responses) [41]. These advancements highlight the rapid evolution of LLMs and their potential.

The integration of LLMs into ophthalmology and corneal disease presents significant potential. Our analysis demonstrated that while LLMs can occasionally match expert-level diagnoses, they exhibit consistent constraints. In highly characteristic presentations such as amiodarone-induced corneal deposits (case #20), with its documented medication history and whorl pattern description, as well as exposure keratopathy (case #9), with clear correlation of lagophthalmos, LLMs from OpenAI, Anthropic, DeepSeek, xAI, and Google achieved diagnostic accuracy comparable to that of human experts. This alignment between LLMs and human corneal specialist extended to the particularly challenging case of Cogan syndrome (case #5), where none of the LLMs generated the correct diagnosis but neither did any of the human experts. There was similar diagnostic reasoning in the assessment of Expert 2 for episcleritis as to that of GPT 4.o in the diagnosis of scleritis and Claude 3.5 Sonnet’s suggestion of anterior scleritis. Similarly, for atopic keratoconjunctivitis (case #3) and corneal marginal ulcer (case #6), diagnostic disagreements were observed between human experts and a majority of the LLMs. Of note, GPT 3.5 correctly identified case #6, whereas the more advanced GPT-4.o models did not, highlighting variability in performance even among iterations of the same LLM family. However, the limitations of LLMs are more pronounced in cases requiring integration of multiple clinical factors. For infectious crystalline keratopathy (case #13) and lattice corneal dystrophy (case #14), all human experts were able to identify the diagnosis, but only one out of the five LLMs succeeded. Case #13 had complex underlying conditions (HSV keratitis, neurotrophic keratitis, and prior penetrating keratoplasty) which could each cause an epithelial defect, and case #14 had masking features, such as Salzmann’s nodules, that distract from the lattice lines and systemic clues with the ptosis and an inability to raise eyebrows. This pattern of diminished performance with complexity was also particularly evident in the ophthalmology-board questions, where the LLMs were limited multistep reasoning and image inference [30,42,43,44]. Notably, almost all the LLMs demonstrated poor performance in diagnosing inflammatory and congenital corneal diseases, with even the best-performing models achieving 60% for inflammatory conditions and most achieving less than 50% for congenital cases. This consistent gap in their pattern recognition may represent inherent biases or limitations in the training data, with these disease categories potentially being underrepresented in the models, especially when compared to their notably better performance with degenerative conditions, for which almost all models had an accuracy ranging from 66.7 to 100%. In healthcare, this becomes especially problematic as biased models can lead to undertreatment of disease or perpetuate detrimental stereotypes by associating certain disease with specific demographic features [45]. These findings emphasize that while LLMs exhibit diagnostic patterns akin to human experts for some corneal cases, they still have significant limits. Such constraints must be carefully considered when these technologies are applied to clinical care, as diagnostic accuracy directly impacts patient outcomes. Further refinement is crucial to bridging these gaps and optimizing LLMs for use in ophthalmology.

Each LLM evaluated in this study demonstrated distinct advantages and limitations. GPT-4.o exhibited the highest diagnostic accuracy and strong concordance with human experts, reflecting its advanced reasoning and expansive training dataset. Claude 3.5 Sonnet also performed well, offering a balance between diagnostic reliability and an emphasis on safe, controlled outputs due to its “Constitutional AI” framework. GPT-3.5 and GPT-4.o mini, although accessible and efficient, showed moderate diagnostic accuracy, highlighting the trade-off between computational simplicity and clinical reliability [46]. Google Gemini 1.5 Flash, despite its multimodal training, underperformed significantly, suggesting that general-purpose capabilities do not necessarily translate into specialized medical expertise [47]. DeepSeek-R1, with its open-source architecture and reinforcement learning enhancements, not only achieved high diagnostic accuracy but also demonstrated cost-effectiveness, making it a viable option for resource-limited settings. Grok 3’s advanced reasoning capabilities and high accuracy further underscore its potential as a valuable tool in ophthalmic diagnostics. The superior performance of GPT-4.o likely reflects its expanded training dataset, enhanced reasoning capabilities, and broader context window compared to earlier models, while Claude 3.5 Sonnet’s strong performance may stem from its curated training approach focused on safety and reliability. Overall, while newer models such as GPT-4.o and Claude 3.5 Sonnet demonstrate substantial promise, further specialization and fine-tuning are needed to optimize LLM performance for complex ophthalmologic diagnoses [28].

While our study was designed to assess LLM performance with clinical cases, the investigation has several important limitations. The relatively small sample size of 20 cases, while providing valuable insights, may not fully represent the broad spectrum of corneal pathologies encountered in clinical practice. Additionally, the case selection process may have introduced bias in terms of disease representation and complexity, particularly for rare conditions where diagnostic challenges are more pronounced. Our methodology of using case reports rather than direct patient encounters excludes an important component of clinical decision-making. Often additional information is not readily available, and the clinician needs to work up the diagnosis by selecting appropriate further imaging or follow up examination. This limitation is especially relevant to conditions where the presentation might evolve over time.

This study evaluated the diagnostic capabilities of publicly accessible LLMs in corneal disease diagnosis, comparing their performance to that of human corneal specialists and highlighting both promising advances and important limitations. While GPT-4.o, Grok3, and Claude 3.5 Sonnet achieved diagnostic accuracy comparable to specialists, significant gaps remain in the diagnosis of more complex corneal pathologies. The marked variability in performance across platforms (30–80% accuracy) underscores the need for careful model selection, particularly for clinical applications. These findings suggest that while LLMs hold potential as adjunct diagnostic tools for common corneal diseases, they cannot replace clinical expertise. Given the rapid evolution of LLMs, future research should focus on developing models with enhanced medical reasoning and ophthalmology-specific training. As these technologies progress, maintaining a balance between innovation and clinical safety will be essential for their responsible integration into ophthalmic practice.

### 4.1. Theoretical Implications

This study contributes to the evolving understanding of LLMs in clinical ophthalmology by highlighting their potential and inherent limitations. The findings suggest that while LLMs such as GPT-4.o and Claude 3.5 Sonnet can approach specialist-level performance in certain corneal disease diagnoses, their diagnostic reasoning is still constrained, particularly in complex or less-represented disease categories. These results support the notion that general-purpose LLMs require further domain-specific adaptation and training to achieve consistent clinical accuracy. Additionally, our study reinforces concerns regarding the impact of model biases and dataset representation on diagnostic reliability, offering important theoretical insights into the development and evaluation of medical AI systems.

### 4.2. Practical Implications

From a practical standpoint, our findings indicate that advanced LLMs may serve as valuable adjunctive tools in ophthalmology, especially in settings with limited access to subspecialty care. GPT-4.o, Grok3, and Claude 3.5 Sonnet, in particular, demonstrated performance levels that could support preliminary triage and diagnostic assistance in common corneal diseases. DeepSeek-R1’s combination of high accuracy and cost-efficiency makes it particularly suitable for deployment in resource-constrained environments. Grok 3’s robust diagnostic capabilities further enhance its applicability in clinical settings, offering reliable support in ophthalmic evaluations. However, the marked variability among models and the poor performance in complex or inflammatory cases underscore the necessity for careful clinical oversight. Integrating LLMs into practice should involve a cautious, evidence-based approach, ensuring that these tools are used to augment, rather than replace, human clinical judgment. Future practical applications will benefit from continuous model evaluation, refinement, and integration with multimodal diagnostic tools.

## 5. Conclusions

This study aimed to evaluate the diagnostic capabilities of several publicly available LLMs compared to human corneal specialists in diagnosing various corneal conditions. Our findings demonstrate that advanced LLMs, particularly GPT-4.o, Claude 3.5 Sonnet, and Grok3 achieved diagnostic accuracy approaching that of human experts in many cases, particularly for degenerative and infectious corneal diseases. However, substantial variability was observed across models, and diagnostic performance was notably lower for complex congenital and inflammatory cases. These results highlight both the promise and the current limitations of LLMs in ophthalmic diagnosis, suggesting that while they can serve as valuable adjunctive tools, human clinical expertise remains indispensable. Moreover, ongoing evaluation of these systems must remain rigorous to ensure patient safety and effectiveness across diverse clinical scenarios. The small sample size may not fully capture the complexity of corneal diseases in clinical practice, and static case reports may limit diagnostic information. Future studies should use larger, real-world datasets, evaluate longitudinal performance prospectively, and explore ophthalmology-specialized LLMs for clinical integration.

## Figures and Tables

**Figure 1 diagnostics-15-01221-f001:**
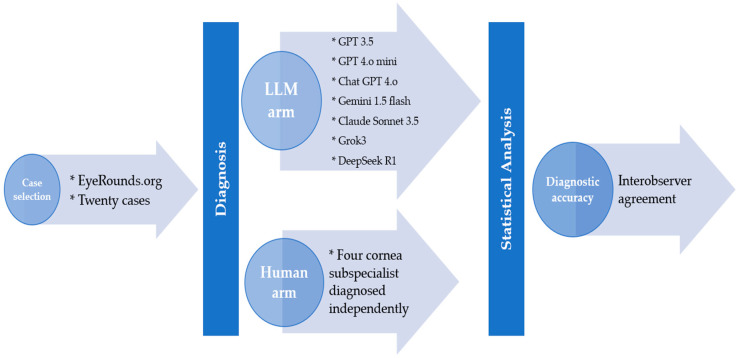
Overview of the methodological workflow.

**Figure 2 diagnostics-15-01221-f002:**
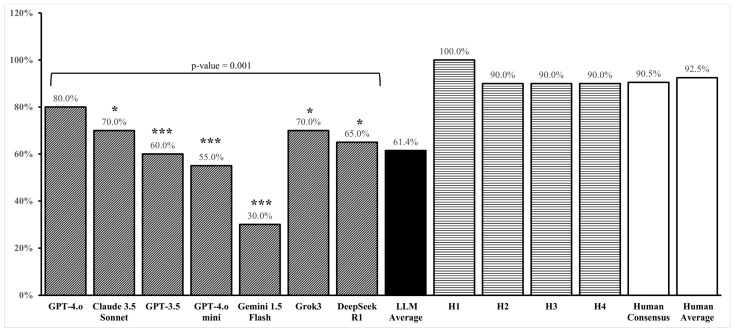
Comparison of the accuracy of various LLMs to that of human cornea specialists denoted with H1, H2, H3, and H4. (*) Indicates significance compared to H1, and (***) indicates significance compared to H1, H2, H3, and H4. *p*-Value indicates Cochran’s Q between all 7 LLMs. No statistically significant difference was observed among the human experts (H1–H4).

**Table 1 diagnostics-15-01221-t001:** Distribution of corneal disease cases by category and demographics.

**Disease Category**	**Number of Cases**	**Percentage**
*Congenital*	6	30%
*Degenerative*	6	30%
*Infectious*	3	15%
*Inflammatory*	5	25%
**Age Group**	**Number of Cases**	**Percentage**
*0–17*	4	20%
*18–35*	2	10%
*36–55*	6	30%
*56–70*	4	20%
*>70*	4	20%
**Gender**	**Number of Cases**	**Percentage**
*Male*	9	45%
*Female*	11	55%

**Table 2 diagnostics-15-01221-t002:** Responses of LLMs and human cornea specialists.

Case	Corneal Eye Disease	Type	GPT 3.5	GPT 4.o Mini	GPT 4.o	Gemini 1.5 Flash	Claude 3.5 Sonnet	Grok3	DeepSeek R1	Human Expert
1	Acanthamoeba Keratitis	Infectious	Acanthamoeba Keratitis	Acanthamoeba Keratitis	Acanthamoeba Keratitis	Acanthamoeba Keratitis	Acanthamoeba Keratitis	Acanthamoeba Keratitis	Acanthamoeba Keratitis	H1: Acanthamoeba Keratitis H2: Acanthamoeba Keratitis H3: Acanthamoeba Keratitis H4: Acanthamoeba Keratitis
2	Acute Corneal Hydrops	Degeneration	Acute Corneal Hydrops	Corneal Hydrops	Acute Hydrops	Keratoconus	Acute Hydrops	Acute Corneal Hydrops	Acute Corneal Hydrops	H1: Acute Corneal Hydrops H2: Acute Corneal Hydrops H3: Acute Corneal Hydrops H4: Acute Corneal Hydrops
3	Atopic Keratoconjunctivitis	Inflammatory	Ocular Cicatricial Pemphigoid	Ocular Cicatricial Pemphigoid	Ocular Cicatricial Pemphigoid	Stevens-Johnson Syndrome	Atopic Keratoconjunctivitis	Ocular Cicatricial Pemphigoid	Ocular Cicatricial Pemphigoid	H1: Atopic Keratoconjunctivitis H2: Atopic Keratoconjunctivitis H3: Ocular Cicatricial Pemphigoid H4: Allergic Keratoconjunctivitis
4	Calcific Band Keratopathy	Degeneration	Superficial Corneal Scar	Band Keratopathy	Band Keratopathy	Schnyder’s Corneal Dystrophy	Calcific Band Keratopathy	Band Keratopathy	Band Keratopathy	H1: Calcific Band Keratopathy H2: Calcific Band Keratopathy H3: Calcific Band Keratopathy H4: Calcific Band Keratopathy
5	Cogan’s Syndrome	Inflammatory	Ocular Rosacea	Peripheral Ulcerative Keratitis	Scleritis	Granulomatous Anterior Uveitis	Anterior Scleritis	Scleritis	Episcleritis	H1: Cogan Syndrome H2: Episcleritis H3: Cogan Syndrome H4: Episcleritis
6	Corneal Marginal Ulcer	Inflammatory	Corneal Marginal Ulcer	Bacterial Keratitis	Herpes Simplex Keratitis	Microbial Keratitis	Peripheral Ulcerative Keratitis	Corneal Ulcer	Microbial Keratitis	H1: Corneal Marginal Ulcer H2: Corneal Marginal Ulcer H3: Mooren Ulcer H4: Corneal Marginal Ulcer
7	Cystinosis	Congenital	Cystinosis	Wilson Disease with Corneal Involvement	Cystinosis	Fabry Disease	Cystinosis	Cystinosis	Cystinosis	H1: Cystinosis H2: Cystinosis H3: Cystinosis H4: Cystinosis
8	Cytarabine Induced Keratoconjunctivitis	Inflammatory	Cytarabine-Induced Keratoconjunctivitis	Cytarabine-Induced Keratopathy	Cytarabine-Induced Keratopathy	Chemotherapy-Induced Dry Eye	Cytarabine-Induced Keratopathy	Cytarabine-Induced Keratoconjunctivitis	Cytarabine-Induced Keratoconjunctivitis	H1: Cytarabine-Induced Keratoconjunctivitis H2: Cytarabine-Induced Keratoconjunctivitis H3: Cytarabine-Induced Keratoconjunctivitis H4: Cytarabine-Induced Keratoconjunctivitis
9	Exposure Keratopathy	Degeneration	Exposure Keratopathy	Exposure Keratopathy	Exposure Keratopathy	Exposure Keratitis	Exposure Keratopathy	Exposure Keratopathy	Exposure Keratopathy	H1: Exposure Keratopathy H2: Exposure Keratopathy H3: Exposure Keratopathy H4: Exposure Keratopathy
10	Fabry Disease	Congenital	Fabry Disease	Fabry Disease	Fabry Disease	Fabry Disease	Fabry Disease	Fabry Disease	Fabry Disease	H1: Fabry Disease H2: Fabry Disease H3: Fabry Disease H4: Fabry Disease
11	Fuchs Endothelial Corneal Dystrophy (FECD)	Congenital	Fuchs Endothelial Corneal Dystrophy	Keratoconus	Fuchs’ Endothelial Corneal Dystrophy	Corneal Endothelial Dystrophy	Posterior Polymorphous Corneal Dystrophy	Fuchs’ Endothelial Corneal Dystrophy	Fuchs’ Endothelial Corneal Dystrophy	H1: FECD H2: FECD H3: FECD H4: FECD
12	Herpes Simplex Virus Keratitis	Infectious	Herpes Simplex Virus Keratitis	Herpes Simplex Keratitis	Herpes Simplex Epithelial Keratitis	Herpes Simplex Virus Keratitis	Herpes Simplex Keratitis	Herpes Simplex Virus Keratitis	Herpes Simplex Virus Keratitis	H1: Herpes Simplex Viral Keratitis H2: Herpes Simplex Viral Keratitis H3: Herpes Simplex Viral Keratitis H4: Herpes Simplex Viral Keratitis
13	Infectious Crystaline Keratopathy	Infectious	Recurrent Herpes Simplex Virus Keratitis	Recurrent Herpes Simplex Virus Keratitis	Infectious Crystaline Keratopathy	Recurrent Herpes Simplex Virus Keratitis	Candida Keratitis	Crystalline Keratopathy	Herpes Simplex Virus Keratitis	H1: ICK H2: ICK H3: ICK H4: ICK
14	Lattice Corneal Dystrophy Type II (Mertetoja’s Syndrome)	Congenital	Meesmann’s Corneal Dystrophy	Anterior Basement Membrane Dystrophy	Lattice Corneal Dystrophy	Meesmann’s Corneal Dystrophy	Oculopharyngeal Muscular Dystrophy	Meesmann’s Corneal Dystrophy	Corneal Dystrophy	H1: Lattice Corneal Dystrophy Type II H2: Lattice Corneal Dystrophy Type II H3: Lattice Corneal Dystrophy Type II H4: Lattice Corneal Dystrophy Type II
15	Megalocornea	Congenital	Positional Pseudo Phacodonesis	Pseudo Phacodonesis with IOL Instability	Pseudo Phacodonesis with IOL Instability	Pupil Dilation with Lens Movement	Megalocornea with Pseudo Phacodonesis	Pseudophacodonesis	Pseudophacodonesis	H1: Megalocornea H2: Megalocornea H3: Megalocornea H4: Megalocornea
16	Peripheral Ulcerative Keratitis	Inflammatory	Peripheral Ulcerative Keratitis	Peripheral Ulcerative Keratitis	Peripheral Ulcerative Keratitis	Neurotrophic Keratitis	Peripheral Ulcerative Keratitis	Peripheral Ulcerative Keratitis	Peripheral Ulcerative Keratitis	H1: PUK H2: PUK H3: PUK H4: PUK
17	Posterior Polymorphous Corneal Dystrophy (PPCD)	Congenital	Congenital Hereditary Endothelial Dystrophy	Posterior Polymorphous Corneal Dystrophy	Posterior Polymorphous Corneal Dystrophy	Guttata Keratopathy	Posterior Stromal Punctate Dystrophy (Posterior Crocodile Shagreen)	Posterior Stromal Corneal Dystrophy	Granular Corneal Dystrophy	H1: PPCD H2: Granular Corneal Dystrophy H3: PPCD H4: PPCD
18	Pseudophakic Bullous Keratopathy	Degeneration	Fuchs Endothelial Corneal Dystrophy	Bullous Keratopathy	Bullous Keratopathy	Corneal Endothelial Dystrophy	Pseudophakic Bullous Keratopathy	Corneal Decompensation Secondary to Endothelial Dysfunction	PBK	H1: PBK H2: PBK H3: PBK H4: PBK
19	Salzmann’s Nodular Degeneration (SND)	Degeneration	Salzmann’s Nodular Degeneration	Pterygium with Secondary Corneal Changes	Salzmann’s Nodular Degeneration	Iron Line Corneal Dystrophy	Salzmann’s Nodular Degeneration	Salzmann Nodular Degeneration	SND	H1: SND H2: SND H3: SND H4: SND
20	Amiodaron-Induced Corneal Deposits (Corneal Verticillata)	Degeneration	Amiodaron-Induced Corneal Deposits	Amiodarone-Related Corneal Deposits	Amiodaron-Induced Corneal Deposits	Amiodaron-Indued Corneal Dystrophy	Amiodarone-Induced Corneal Verticillata	Amiodarone-Induced Corneal Deposits	Amiodarone-Induced Corneal Deposits	H1: Amiodarone-induced corneal deposits H2: Amiodarone-induced corneal deposits H3: Amiodarone-induced corneal deposits H4: Amiodarone-induced corneal deposits

In the table, red cells indicate incorrect diagnoses.

**Table 3 diagnostics-15-01221-t003:** Agreement between LLMs and human cornea specialists.

	GPT 4.o	Claude 3.5 Sonnet	GPT 3.5	GPT 4.o Mini	Gemini 1.5 Flash	Grok3	DeepSeek R1
** *Interobserver Agreement H1* **	80% (16/20)	60% (12/20)	60% (12/20)	55% (11/20)	80% (16/20)	60% (12/20)	65% (13/20)
** *Interobserver Agreement H2* **	75% (15/20)	60% (12/20)	60% (12/20)	50% (10/20)	75% (15/20)	65% (13/20)	75% (15/20)
** *Interobserver Agreement H3* **	85% (17/20)	60% (12/20)	60% (12/20)	60% (12/20)	85% (17/20)	65% (13/20)	70% (14/20)
** *Interobserver Agreement H4* **	80% (16/20)	65% (13/20)	60% (12/20)	65% (13/20)	80% (16/20)	65% (13/20)	70% (14/20)
** *Kappa Agreement with Human Expert Consensus* **	0.348	0.219	0.146	0.121	0.044	0.219	0.178
***Kappa p*-*Value***	**0.04**	0.117	0.209	0.257	0.502	0.117	0.162

**Table 4 diagnostics-15-01221-t004:** LLM performance by corneal disease category.

Model	Degenerative	Inflammatory	Congenital	Infectious
** *GPT 4.o* **	100.0% (6/6)	40.0% (2/5)	83.3% (5/6)	100.0% (3/3)
** *Claude 3.5 Sonnet* **	100.0% (6/6)	60.0% (3/5)	50.0% (3/6)	66.7% (2/3)
** *GPT 3.5* **	66.7% (4/6)	60.0% (3/5)	50.0% (3/6)	66.7% (2/3)
** *GPT 4.o mini* **	83.3% (5/6)	40.0% (2/5)	33.3 (2/6)	66.7% (2/3)
** *Gemini 1.5 Flash* **	33.3% (2/6)	0.0% (0/5)	33.3% (2/6)	66.7% (2/3)
** *Grok3* **	83.3% (5/6)	40% (2/5)	66.7% (4/6)	100% (3/3)
** *DeepSeek R1* **	100% (6/6)	40% (2/5)	50% (3/6)	66.7% (2/3)

**Table 5 diagnostics-15-01221-t005:** Summary of recent studies evaluating LLMs in ophthalmology.

Author (Year)	Study Focus	LLMs Evaluated	Key Findings
Antaki et al. [20]	ChatGPT performance on OKAP-style questions	ChatGPT Legacy, ChatGPT Plus	Moderate accuracy (42.7–59.4%), performance influenced by question difficulty and subspecialty.
Chen et al. [28]	ChatGPT performance on ophthalmic case assessments	ChatGPT	88.2% (15 of 17) diagnostic accuracy.
Mihalache et al. [29]	Gemini and Bard performance on board questions	Google Gemini, Bard	71% accuracy (across 150 text-based multiple-choice questions); minor country-based performance variability.
Cai et al. [30]	Generative LLMs on board-style questions	ChatGPT-3.5, ChatGPT-4.0, Bing Chat	GPT-4.0 (71.6%) and Bing Chat (71.2%) near human performance (72.2%); hallucination rates: GPT-3.5 (42.4%) > Bing Chat (25.6%) > GPT-4.0 (18.0%).
Delsoz et al. [22]	LLM diagnosis of corneal diseases	ChatGPT-3.5, ChatGPT-4.0	GPT-4.0 achieved 85% (17 of 20 cases) vs. 60% (12 of 20 cases) for GPT-3.5; high agreement of GPT-4.0 with cornea specialists.
Taloni et al. [21]	Performance on AAO self-assessment questions	ChatGPT-3.5, ChatGPT-4.0	GPT-4.0 outperformed humans (82.4% vs. 75.7%), and GPT-3.5 had less accurate answers (65.9%). Both GPT-4.0 and GPT-3.5 showed the worst results in surgery-related questions (74.6% and 57.0%, respectively)
Bernstein et al. [31]	Comparison of LLM vs. ophthalmologists on patient question and answer	ChatGPT	Similar quality, appropriateness, and safety in AI and human responses—difficult to distinguish by experts.
Holmes et al. [32]	LLMs in pediatric ophthalmology education	ChatGPT-3.5, GPT-4, PaLM2	GPT-4 matched attending physicians; GPT-3.5 > students; GPT-4 showed most consistency/confidence.
Huang et al. [33]	LLM performance in glaucoma and retina management	GPT-4	GPT-4 outperformed glaucoma specialists and matched retina experts in diagnostic and treatment accuracy.
Kreso et al. [34]	LLMs in emergency ophthalmology	GPT-4, GPT-4o, Llama-3-70b	GPT-4 (score = 3.52) and Llama-3-70b (score = 3.48) performed similarly to experts (score = 3.72); GPT-4o underperformed.

## Data Availability

Data supporting the findings of this study are available from the corresponding author upon reasonable request.

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
