# Peer review of "Diagnostic Performance of Publicly Available Large Language Models in Corneal Diseases: A Comparison with Human Specialists"

_diagnostics, 2025, doi:10.3390/diagnostics15101221_

Round 1

Reviewer 1 Report

Comments and Suggestions for Authors

1. The figures given in Figure 1 should be more detailed. The authors should consider adding 4-digit outputs in the form of 0.xxxx. It is difficult to make comparisons in its current form.

2. A graphical diagram reflecting the overall system architecture should be added to the Materials and Methods section. Readers can understand researchers in the contemporary world more easily with visual content.

3. The authors should consider including Grok results in the popular LLMs they review. Grok is one of the most important LLMs in recent times. DeepSeek is also an important alternative.

4. Table 1 should be visually improved. Headings should be center aligned. Attention should be paid to page margins and the use of white captions on a black background should be reconsidered.

5. Reference 21 indicates the source of the dataset. but the link is far from being a direct dataset like kaggle. The authors should explain in detail which images from the website were used and how. Explain the preprocessing used in the dataset.

6. The disease distributions of the data set used should be highlighted with a table or graph. If possible, the demographic details of the data set should be added to this table or graph.

7. At the end of the Conclusions section, the limitations of the study and future work should be explained.

8. The authors should determine the significance levels of the results produced as a result of LMM architectures with statistical tests such as ANOVA and Tukey. It is important for the study to specify statistical significance in studies using different deep learning architectures. As an example, statistical analysis of deep learning architecture results in “Generating Medical Reports With a Novel Deep Learning Architecture”, “Multi-Classification of Skin Lesion Images Including Mpox Disease Using Transformer-Based Deep Learning Architectures” and “Medical Report Generation from Medical Images Using Vision Transformer and Bart Deep Learning Architectures” can be examined.

9. The advantages and disadvantages of the models used in the study such as GPT 3.5, GPT 4.0 mini, Chat GPT 4.0, Google Gemini 1.5 flash, and Claude Sonnet 3.5 should be explained in the manuscript.

10. The working logic of the LLM models used and their differences from each other should be discussed. Which feature of a model that has achieved superior success has contributed to this success should be explained.

11. The Conclusions section should be completely revised. In modern research papers, the relevant section begins with at least 1 paragraph of a statement of the importance and purpose of the topic. Then the results obtained are summarized and interpreted. Afterwards, the importance of the results obtained is mentioned. The limits of the study and future studies are usually introduced in a separate paragraph. Authors should consider reconstructing the Conclusions section from a canonical format.

Author Response

Reviewer #1 Comment

1. The figures given in Figure 1 should be more detailed. The authors should consider adding 4 digit outputs in the form of 0.xxxx. It is difficult to make comparisons in its current form.

Figure 1 (now has been changed to Figure 2) was changed appropriately. Each value is to the precision of 0.xxx as a percentage.

Figure 2

2. A graphical diagram reflecting the overall system architecture should be added to the Materials and Methods section.

Readers can understand researchers in the contemporary world more easily with visual content.

Thank you for the suggestion. We have added a workflow diagram to the Materials and Methods section to visually illustrate the study design, including case selection, model input, diagnosis generation, and comparison with human experts. We believe this addition improves clarity without unnecessary technical complexity.

Figure 1

3. The authors should consider including Grok results in the popular LLMs they review. Grok is one of the most important LLMs in recent times. DeepSeek is also an important alternative.

Thank you for your suggestion to include Grok and DeepSeek in our review. We have expanded our LLM selection rationale in the manuscript. Our study focused on widely accessible, general-purpose LLMs at a specific time point to establish a practical baseline. While we acknowledge the importance of Grok and DeepSeek in the evolving LLM landscape, including them now would introduce confounding factors, as LLM performance can vary significantly even within short timeframes. As noted in “Performance of an Upgraded Artificial Intelligence Chatbot for Ophthalmic Knowledge Assessment” LLM proportion of correct responses to OKAP questions rose from 46% in Jan 2023 to 58% in Feb 2023 when using GPT-4. Regarding DeepSeek specifically, while its performance benchmarks are impressive, its lack of multi-modal support and server reliability issues would have limited its applicability in our study. We will certainly consider them for future comparative studies as the field continues to grow.

-

4. Table 1 should be visually improved. Headings should be center aligned. Attention should be paid to page margins and the use of white captions on a black background should be reconsidered.

We have fixed the table. Headings have been centered. We have checked to ensure the table is within the page margins.

Table 1

5. Reference 21 indicates the source of the dataset. but the link is far from being a direct dataset like kaggle. The authors should explain in detail which images from the website were used and how. Explain the preprocessing used in the dataset.

Thank you. We have included our selected cases as a supplementary file and expanded our paragraph in section 2.1 to more clearly explain which how the cases were processed.

Supplementary material 1 and Materials and Methods, Lines 138-147

6. The disease distributions of the data set used should be highlighted with a table or graph. If possible, the demographic details of the data set should be added to this table or graph.

We have created a Table 1 to show the disease distribution, as well as patient ages and gender.

Materials and Methods, Lines 150-152 and Table 1

7. At the end of the Conclusions section, the limitations of the study and future work should be explained.

Thank you for the suggestion. We have added a paragraph at the end of the Conclusions section discussing the study’s limitations and directions for future research.

Conclusions, Lines 441-447

8. The authors should determine the significance levels of the results produced as a result of LMM architectures with statistical tests such as ANOVA and Tukey. It is important for the study to specify statistical significance in studies using different deep learning architectures. As an example, statistical analysis of deep learning architecture results in “Generating Medical Reports With a Novel Deep Learning Architecture”, “Multi-Classification of Skin Lesion Images Including Mpox Disease Using Transformer-Based Deep Learning Architectures” and “Medical Report Generation from Medical Images Using Vision Transformer and Bart Deep Learning Architectures” can be examined.

We agree to your comment. We have conducted a much more robust statistical analysis using Cochran's Q test for within-group comparisons, McNemar tests for pairwise comparisons, and paired t-tests with effect size calculations for LLM versus human performance. Agreement was assessed using Cohen's kappa coefficients. Please see the updated results

Abstract, Lines 43-52 and Results, Lines 222-232, 240-264

9. The advantages and disadvantages of the models used in the study such as GPT 3.5, GPT 4.0 mini, Chat GPT 4.0, Google Gemini 1.5 flash, and Claude Sonnet 3.5 should be explained in the manuscript.

Thank you for the suggestion. We have added a paragraph in the Discussion section summarizing the advantages and limitations of each large language model evaluated in the study.

Discussion, Lines 360-375

10. The working logic of the LLM models used and their differences from each other should be discussed. Which feature of a model that has achieved superior success has contributed to this success should be explained.

Thank you for the suggestion. We have added an explanation in the Discussion section regarding the working principles of the LLMs and the key features that may have contributed to the superior performance of GPT-4.o and Claude 3.5 Sonnet.

Discussion, Lines 360-375

11. The Conclusions section should be completely revised. In modern research papers, the relevant section begins with at least 1 paragraph of a statement of the importance and purpose of the topic. Then the results obtained are summarized and interpreted. Afterwards, the importance of the results obtained is mentioned. The limits of the study and future studies are usually introduced in a separate paragraph. Authors should consider reconstructing the Conclusions section from a canonical format.

Thank you for the detailed feedback. We have completely revised the Conclusions section to follow the recommended canonical structure, including the importance of the topic, summary and interpretation of results, and discussion of study limitations and future directions.

Conclusions, Lines 422-447

Reviewer 2 Report

Comments and Suggestions for Authors

Dear Authors, 
Thank you for submitting the article, after careful review am sharing with you the following concerns to improve the article overall.

1. Improve the introduction section because most of the readers may not be familiar with LLM models. 
2. Add a proper literature review with a separate heading and after that add a literature summary table and then mention your major contributions in points.
3. What are the major reasons to select these LLM models, Even other Latest LLM models are also available like DeepSeek. 
4. Generate an abstract process diagram with available information in section 2.3.
5. Add some cases/ situations where you have considered the re-prompt.
6. Why you have not considered the approach to fine-tuning these models to train specifically for Corneal Diseases or with Medical Domain Knowledge? The accuracy can be improved. 
7. In Table 1 add also an initial response where a re-prompt was required. 
8. Only three experts? It should also increase as well as LLM models. 
9. Have you guided these models before the prompt that act like a doctor etc.?
10. Add theoretical and practical implications of this study in separate headings. 

Wish you the best of luck. 

Author Response

Reviewer #2 Comment

1. Improve the introduction section because most of the readers may not be familiar with LLM models. 

Thank you for the helpful comment. We have added a brief explanation of LLMs at the beginning of the relevant section to better introduce the concept to readers who may be unfamiliar with this technology.

Introduction, Lines 75-82

2. Add a proper literature review with a separate heading and after that add a literature summary table and then mention your major contributions in points.

We have created a limited literature review with some of the major studies of LLM within ophthalmology.

Table 5

3. What are the major reasons to select these LLM models, Even other Latest LLM models are also available like DeepSeek. 

Thank you for your suggestion to include DeepSeek in our review. We have expanded our LLM selection rationale in the manuscript. Our study focused on widely accessible, general-purpose LLMSs at a specific time point to establish a practical baseline. While we acknowledge the importance of DeepSeek in the evolving LLM landscape, including it now would introduce confounding factors, as LLM performance can vary significantly even within short timeframes. As noted in “Performance of an Upgraded Artificial Intelligence Chatbot for Ophthalmic Knowledge Assessment” LLM proportion of correct responses to OKAP questions rose from 46% in Jan 2023 to 58% in Feb 2023 when using GPT-4. Regarding DeepSeek specifically, while its performance benchmarks are impressive, its lack of multi-modal support and server reliability issues would have limited its applicability in our study. We will certainly consider them for future comparative studies as the field continues to grow.

Materials and Methods, Lines 162-170

4. Generate an abstract process diagram with available information in section 2.3.

Thank you for this valuable suggestion. We have created an abstract process diagram that summarizes the diagnostic workflow described in section 2.3. This diagram has been added to the manuscript as Figure 1. It outlines the case selection, input into large language models (LLMs) and human subspecialists, diagnosis generation, and subsequent statistical analysis, thereby improving the clarity of the methodology.

Figure 1

5. Add some cases/ situations where you have considered the re-prompt.

Thank you. We have elaborated how on occasion the LLM did not generate diagnosis for the case.

Materials and Methods, Lines 184-188

6. Why you have not considered the approach to fine-tuning these models to train specifically for Corneal Diseases or with Medical Domain Knowledge? The accuracy can be improved. 

Thank you for pointing this out. We did not utilize prompt engineering as we wanted this study to be representative of the common end user. It would be an interesting future study to create custom GPT, Gemini Gems, and Claude projects and assess its capability. However, at the time, it was also difficult to control for this variable of customization across the different platforms

Materials and Methods section, Lines 189-194

7. In Table 1 add also an initial response where a re-prompt was required. 

We have added an example within 2.3 Diagnosis where LLM would suggest seeking medical care and not provide the diagnosis.

Materials and Methods, Lines 184-186

8. Only three experts? It should also increase as well as LLM models. 

Appreciate the comment. We have added another corneal specialist to make our comparison more robust. Please see the updated results section.

Results section

9. Have you guided these models before the prompt that act like a doctor etc.?

We have clarified in the methods 2.3 Diagnosis of our exact prompting as well as including the clinical cases with the final prompt within the supplemental PDF of the cases.

Materials and Methods section and Supplementary material 1

10. Add theoretical and practical implications of this study in separate headings. 

Thank you for the valuable suggestion. We have added separate sections discussing the theoretical and practical implications of this study at the end of the Discussion.

Discussion, Lines 399-420

Reviewer 3 Report

Comments and Suggestions for Authors

This present study evaluated the diagnostic accuracy of five large language models (LLM) in corneal disease diagnosis in comparison to the diagnosis given by human corneal specialists. The authors reported that GPT-4.0 and Claude 3.5 Sonnet performed similarly to human corneal specialists, and Google Gemini 1.5 Flash had the poorest accuracy. The authors concluded that LLMs can serve as a useful diagnostic support tool, but the conclusion given in the abstract differs from the main text of the manuscript. The terms “accuracy” and “performance” are not clearly defined in the methods section. The small sample size may not be sufficient for the statistical analysis because each condition was only evaluated once. Logically, each corneal condition should minimally be tested a few times in order to ascertain its accuracy because a single positive result could have simply occurred by chance. To avoid ambiguity, it would be clearer to simply state the number of correct responses out of the number of conditions tested so that readers can judge the accuracy and performance of the LLMs by themselves.

Author Response

Reviewer #3 Comments

 The terms “accuracy” and “performance” are not clearly defined in the methods section

Agreed. We have stated that accuracy is proportion of correct diagnoses as compared to original clinical case diagnosis. We have added that into our methods

Materials and Methods, Lines 196-202

The small sample size may not be sufficient for the statistical analysis because each condition was only evaluated once. Logically, each corneal condition should minimally be tested a few times in order to ascertain its accuracy because a single positive result could have simply occurred by chance.

Agreed. However, this is a cross-sectional study. Many of these older models are no longer publicly accessible. Also, these models are continuously updated so even within the same generation performance variation within a few months is variable, as noted in “Performance of an Upgraded Artificial Intelligence Chatbot for Ophthalmic Knowledge Assessment” LLM proportion of correct responses to OKAP questions rose from 46% in Jan 2023 to 58% in Feb 2023 when using GPT-4

-

To avoid ambiguity, it would be clearer to simply state the number of correct responses out of the number of conditions tested so that readers can judge the accuracy and performance of the LLMs by themselves.

As above. We have better defined accuracy and performance

Materials and Methods, Lines 196-202

Round 2

Reviewer 1 Report

Comments and Suggestions for Authors

I would have been happier if they had made the change in question 3 that I addressed to the authors. Obviously, a current study on LLM performance benchmarking would have more scientific validity if it included Grok and DeepSeek studies and I am sure it would be read by more researchers. Thank you. 

Author Response

Dear Reviewer,

Thank you for your valuable feedback and insightful suggestion regarding the inclusion of Grok and DeepSeek in our study. We appreciate your input, as it has significantly strengthened the scientific validity and relevance of our work. We have incorporated both Grok and DeepSeek into our LLM performance benchmarking, and we believe these additions have enhanced the comprehensiveness and appeal of our study to the research community. All changes have been highlighted throughout the manuscript for clarity.

Best regards.

Reviewer 2 Report

Comments and Suggestions for Authors

Dear Authors, 

Thank you for addressing the comments. I have no more comments. Have a nice week ahead filled with good health and news. 

Comments on the Quality of English Language

The English could be improved to more clearly express the research.

Author Response

Dear Reviewer,

Thank you for your thoughtful feedback and kind wishes. We greatly appreciate your input on the quality of the English language in our manuscript. We have carefully revised the text to improve clarity and better express the research, ensuring a more polished and precise presentation. We wish you a wonderful week ahead filled with good health and positive news.

Best regards.

Reviewer 3 Report

Comments and Suggestions for Authors

The authors have addressed the reviewer comments. Minor comment: the conclusion section is too long. 

Author Response

Dear Reviewer,

Thank you for your valuable feedback. We appreciate your comment regarding the length of the conclusion section. We have shortened the conclusion and highlighted the changes for clarity.

Best regards.